# Clinical and Economic Impact of Third-Generation Cephalosporin-Resistant Infection or Colonization Caused by *Escherichia coli* and *Klebsiella pneumoniae*: A Multicenter Study in China

**DOI:** 10.3390/ijerph17249285

**Published:** 2020-12-11

**Authors:** Xuemei Zhen, Cecilia Stålsby Lundborg, Xueshan Sun, Xiaoqian Hu, Hengjin Dong

**Affiliations:** 1Centre for Health Management and Policy Research, School of Public Health, Cheeloo College of Medicine, Shandong University, (National Health Commission (NHC) Key Laboratory of Health Economics and Policy Research, Shandong University), Jinan 250012, China; zhenxuemei@sdu.edu.cn; 2Center for Health Policy Studies, School of Public Health, Zhejiang University School of Medicine, Hangzhou 310058, China; sunxueshan@zju.edu.cn (X.S.); huxiaoqian@qdu.edu.cn (X.H.); 3Department of Global Public Health, Karolinska Institutet, 17177 Stockholm, Sweden; Cecilia.Stalsby.Lundborg@ki.se; 4College of Politics and Public Administration, Qingdao University, Qingdao 266061, China; 5The Fourth Affiliated Hospital Zhejiang University School of Medicine, No. N1, Shancheng Avenue, Yiwu 322000, China

**Keywords:** *Escherichia coli*, *Klebsiella pneumoniae*, third-generation cephalosporin, 3GCREC, 3GCRKP, economic cost, length of stay, hospital mortality

## Abstract

Quantifying economic and clinical outcomes for interventions could help to reduce third-generation cephalosporin resistance and *Escherichia coli* or *Klebsiella pneumoniae*. We aimed to compare the differences in clinical and economic burden between third-generation cephalosporin-resistant *E. coli* (3GCREC) and third-generation cephalosporin-susceptible *E. coli* (3GCSEC) cases, and between third-generation cephalosporin-resistant *K. pneumoniae* (3GCRKP) and third-generation cephalosporin-susceptible *K. pneumoniae* (3GCSKP) cases. A retrospective and multicenter study was conducted. We collected data from electronic medical records for patients who had clinical samples positive for *E. coli* or *K. pneumoniae* isolates during 2013 and 2015. Propensity score matching (PSM) was conducted to minimize the impact of potential confounding variables, including age, sex, insurance, number of diagnoses, Charlson comorbidity index, admission to intensive care unit, surgery, and comorbidities. We also repeated the PSM including length of stay (LOS) before culture. The main indicators included economic costs, LOS and hospital mortality. The proportions of 3GCREC and 3GCRKP in the sampled hospitals were 44.3% and 32.5%, respectively. In the two PSM methods, 1804 pairs and 1521 pairs were generated, and 1815 pairs and 1617 pairs were obtained, respectively. Compared with susceptible cases, those with 3GCREC and 3GCRKP were associated with significantly increased total hospital cost and excess LOS. Inpatients with 3GCRKP were significantly associated with higher hospital mortality compared with 3GCSKP cases, however, there was no significant difference between 3GCREC and 3GCSEC cases. Cost reduction and outcome improvement could be achieved through a preventative approach in terms of both antimicrobial stewardship and preventing the transmission of organisms.

## 1. Introduction

*Escherichia coli* and *Klebsiella pneumoniae*, both species of the family Enterobacteriaceae, are the most prevalent gram-negative bacteria causing intra-abdominal infection, urinary tract infection, and bloodstream infection [1,2], and can be resistant to the widely used antibiotics, such as third-generation cephalosporins, namely third-generation cephalosporin-resistant *E. coli* (3GCREC) and third-generation cephalosporin-resistant *K. pneumoniae* (3GCRKP) [3,4]. The World Health Organization (WHO) classified 3GCREC and 3GCRKP as critical-priority bacteria [5]. Alvarez-Uria et al. (2018) pointed out that global resistant prevalence was 64.5% for 3GCREC and 66.9% for 3GCRKP by 2030 [6]. The China Antimicrobial Resistance Surveillance System reported that the average proportion of 3GCREC and 3GCRKP in 2019 was 51.9% and 31.9%, respectively [7], which was higher than the levels in United Kingdom (11.0% and 13.0%) and in Sweden (8.3% and 5.5%) [8].

Third-generation cephalosporin resistance in *E. coli* or *K. pneumoniae* is a global concern [9,10]. Infections caused by 3GCREC and 3GCRKP were associated with higher mortality, longer length of stay (LOS), and more economic costs compared with susceptible cases [11,12,13]. de Kraker et al. (2011) showed that 15,183 episodes of 3GCREC were associated with 2712 excess deaths, 120,065 extra LOS, and €18.1 million increased costs in 31 European countries [13]. It was concluded that patients with third-generation cephalosporin-resistant Enterobacteriaceae contributed to 16.1% of hospital mortality, 4.9 days of LOS, and €320 of infection cost in one study by Stewardson et al. (2016) [14]. In addition, colonization of *E. coli* and *K. pneumoniae*, as the reservoir for infection with these organisms, was also a risk factor for higher mortality, longer LOS, and increased hospital costs [15,16].

Quantifying clinical and economic outcomes would facilitate strategies towards the containment of third-generation cephalosporin resistance and *E. coli* or *K. pneumoniae*. Resistance to third-generation cephalosporins by *E. coli* or *K. pneumoniae*, which represented the major mechanism of antimicrobial resistance, had been reported as independently associated with a poor outcome and increased use of healthcare resources [12,17]. However, no significant difference in hospital mortality between 3GCREC and third-generation cephalosporin-susceptible *E. coli* (3GCSEC) was reported [13,18]. In China, there was only one study exploring longer LOS and higher hospital costs attributable to extended spectrum beta-lactamase (ESBL)-positive intra-abdominal infection caused by *E. coli* or *K. pneumoniae* [19]. The clinical and economic outcomes of 3GCREC and 3GCRKP remained largely uninvestigated in China. In this study, we aimed to compare the clinical and economic difference between 3GCREC and 3GCSEC, and between 3GCRKP and third-generation cephalosporin-susceptible *K. pneumoniae* (3GCSKP), in China.

## 2. Materials and Methods

### 2.1. Study Site

We conducted this study in four tertiary hospitals in China; three in Zhejiang Province (Site 1, Site 3, and Site 4) are a general provincial hospital, general county hospital, and combined traditional Chinese and Western medicine provincial hospital, respectively, and one in Shandong Province (Site 2) is a general provincial hospital. There are 3200, 3500, 1727, 2100 of hospital beds and 170,000, 160,000, 80,000, 50,000 inpatients per year in these four hospitals, respectively.

### 2.2. Study Design and Patients

A retrospective and multicenter study was conducted. We collected data from electronic medical records (EMR) for patients who had clinical samples positive for *E. coli* or *K. pneumoniae* isolates, that were detected in any specimens (e.g., blood, stool, cervical, and urethral sources) between 2013 and 2015 [20]. Patients were defined as 3GCREC/3GCRKP cases if patients infected or colonized by *E. coli* or *K. pneumoniae* were resistant or intermediate to any third-generation cephalosporin or as 3GCSEC/3GCSKP cases if they were susceptible to all third-generation cephalosporins according to the Clinical and Laboratory Standards Institute (CLSI) definitions [15,21]. We only included the first episode for each patient to avoid duplication. The study was approved by the institutional review board of Zhejiang University School of Public Health, who waived the need for informed consent. All inpatients data were anonymized prior to analysis.

### 2.3. Data Collection

We collected patient characteristics from EMR. The data for each patient included demographics (age, sex, and insurance), comorbidities (disease diagnosis, and Charlson comorbidity index (CCI), hospital events (admitting service, surgical services, and date of hospital and intensive care unit (ICU) admission or discharge), microbiological data, clinical outcomes (discharged alive or death during hospitalization), and economic costs.

### 2.4. Propensity Score Matching

To minimize the impact of potential confounding variables, we performed propensity score matching (PSM) with 1:1 nearest-neighbor matching. PSM, widely used to control for confounding in observational studies, is a powerful statistical matching technique for reducing a set of confounding variables to a single propensity score in order to effectively control for all observed confounding bias [22]. There were two step-by-step rounds of PSM. First, we employed a logistic regression model with third-generation cephalosporin-resistant or-susceptible as dependent variables, and with age, sex, insurance, number of diagnoses, CCI, admission to ICU, surgery, and comorbidities as independent variables. Second, because LOS is the major contributor to additional economic cost, we repeated the PSM including LOS before culture as a potential confounding variable. The generated pairs matched with potential confounding variables were subjected to further analyses of economic costs, LOS and hospital mortality.

### 2.5. Indicators and Statistical Analyses

The main indicators included economic costs, LOS and hospital mortality. The economic costs comprised total hospital cost, medication cost (antibiotic cost), diagnostic cost, treatment cost, material cost, and other costs, and they covered out-of-pocket payment by patients themselves and payments by health insurers. All economic costs were presented in 2015 United States (US) dollars values according to purchasing power parities and the consumer price index of China [23,24].

The Wilcoxon rank-sum test and χ^2^ test were conducted to compare the main indicators between 3GCREC and 3GCSEC and between 3GCRKP and 3GCSKP for the quantitative and qualitative variables, respectively. Statistical analyses were performed using STATA. All *p*-values were two-tailed, and those less than 0.05 were considered statistically significant.

## 3. Results

The proportions of 3GCREC and 3GCRKP in the sampled hospitals were 44.3% and 32.5%, respectively. A total of 2056 inpatients infected or colonized with 3GCREC and 2588 with 3GCSEC, 1679 with 3GCRKP and 3485 with 3GCSKP were included during the study period. There were significant differences in sex, admission to ICU, surgery, and some comorbidities between the 3GCREC and 3GCSEC groups, and in age, number of diagnoses, admission to ICU, surgery, and some comorbidities between the 3GCRKP and 3GCSKP groups before PSM. Therefore, we conducted PSM to minimize the influencing of variables in two steps. First, excluding LOS before culture as a potential confounding variable, we obtained 1815 pairs and 1617 pairs, respectively. In addition, 1804 pairs and 1521 pairs were generated, respectively, after PSM for potential confounding variables including LOS before culture. There were no differences in patients’ characteristics between the two groups after PSM (Table 1).

After PSM for potential confounding variables excluding LOS before culture, inpatients with third-generation cephalosporin resistance were significantly associated with higher economic costs and LOS than susceptible cases. The median differences (95% certainty interval (CI)) in total hospital cost, antibiotic cost, medication cost, diagnostic cost, treatment cost, and material cost were $1366 ($1179–$1453), $152 ($146–$168), $627 ($577–$715), $81 ($57–$79), $363 ($324–$393), and $134 ($129–$143), respectively, for inpatients with 3GCREC (Table 2), and were $7671 ($7419–$7932), $881 ($809–$982), $4461 ($4168–$4658), $620 ($566–$708), $1612 ($1501–$1756), and $583 ($535–$641), respectively, for inpatients with 3GCRKP (Table 3). The median LOS of inpatients with 3GCREC and 3GCRKP were longer than those with 3GCSEC and 3GCSKP, with a difference of 4 days and 11 days, respectively (Table 4). In addition, there was no significant difference in hospital mortality between the 3GCREC and 3GCSEC groups (*p* = 0.281), however, a significant difference with 3.09% (2.78–3.39%) of hospital mortality was found between the 3GCRKP and 3GCSKP groups (*p* < 0.000) (Table 5).

After PSM for potential confounding variables including LOS before culture, the differences in economic costs, LOS and hospital mortality for inpatients with 3GCREC and 3GCRKP were lower than the results after PSM for variables excluding LOS before culture. The differences in total hospital cost, antibiotic cost, medication cost, diagnostic cost, treatment cost, and material cost between the 3GCREC and 3GCSEC groups and between the 3GCRKP and 3GCSKP groups were statistically significant, with median differences of $1140 ($942–$1227), $127 ($127–$147), $515 ($456–$592), $67 ($61–$85), $271 ($245–$296), and $107 ($101–$114), respectively, for inpatients with 3GCREC (Table 2), and with median differences of $4763 ($4340–$5024), $729 ($655–$814), $2998 ($2695–$3310), $445 ($380–$460), $952 ($989–$1015), and $340 ($299–$383), respectively, for inpatients with 3GCRKP (Table 3). The LOS of inpatients with 3GCREC or 3GCRKP was significantly longer than that of inpatients with 3GCSEC or 3GCSKP, with a median difference of 2.5 days and 7 days, respectively (Table 4). In addition, no significant difference in hospital mortality between the 3GCREC and 3GCSEC groups was found (*p* = 0.508), but significant difference existed between the 3GCRKP and 3GCSKP groups (*p* = 0.001) (Table 5).

## 4. Discussion

Previous studies mainly focused on antibiotic utilization and resistance mechanisms and the clinical and economic outcomes of 3GCREC and 3GCRKP in China remained largely uninvestigated. To the best of our knowledge, this is the first study to quantify the clinical and economic outcome of 3GCREC and 3GCRKP in mainland China using the PSM method with large sample size and multiple hospital settings. We focused on *E. coli* or *K. pneumoniae*, avoiding non-specific effects from a combination of bacteria [14,25]. In this study, we found that compared with third-generation cephalosporin-susceptible cases, those with 3GCREC and 3GCRKP were associated with significantly increased total hospital cost and excess LOS. In addition, inpatients with 3GCRKP were significantly associated with higher hospital mortality compared with 3GCSKP cases, however, there was no significant difference between the 3GCREC and 3GCSEC groups.

Conducting economic and clinical evaluation for interventions could help to reduce the transmission of 3GCREC or 3GCRKP in hospital settings [26]. It was demonstrated that third-generation cephalosporin resistance increased the economic costs and prolonged the LOS among inpatients with *E. coli* and *K. pneumoniae* [11,12,13,14,15,18,19,25,27,28,29,30,31,32]. For example, Hu et al. (2010) showed that ESBL-positive intra-abdominal infection led to attributable hospital costs and excess hospital stay in China [19]. MacVane et al. (2018) reported that urinary tract infection caused by ESBL-producing *E. coli* or *K. pneumoniae* was associated with significant hospital cost and hospital stay in the United States [31]. Meanwhile, one study explored the possibility that colonization with ESBL producing *E. coli* was associated with longer LOS and higher hospital costs as well [16].

In addition, inpatients with 3GCRKP were significantly associated with higher hospital mortality compared with those with 3GCSKP, which was consistent with other studies [2,32]. However, there was no significant difference in hospital mortality between 3GCREC and 3GCSEC in our study, which was different compared to other studies conducted in European countries [13,18]. Meanwhile, some studies also found there was no difference in hospital mortality between ESBL-producing *E. coli* cases and non-ESBL-producing cases [16,33]. Different conclusions might be associated with different study design, sample size, geography, resistant pattern, etc. Therefore, this finding needs to be further explored in the future. In addition, the manners in which the use of beta-lactams might affect prevalence of third-generation cephalosporin resistance remained to be fully elucidated [27].

LOS could increase daily bed cost, and might contribute to more treatment service and diagnostic service, therefore, LOS was the major contributor to economic costs [34]. In this study, we applied the PSM method using two step-by-step rounds [29,30,35]. Although the inclusion of LOS before culture as an independent variable in PSM could attenuate the effect of 3GCREC or 3GCRKP on economic costs, LOS and hospital mortality, the conclusion was unchanged when LOS before culture was excluded between the two groups.

This study is not without limitations. First, due to the retrospective nature of our study, it was difficult to distinguish infection or colonization. It was necessary to explore the burden of 3GCREC and 3GCRKP, either infection or colonization, because colonization was an important reservoir for organisms of infection. Prospective studies among patients with infections need to be conducted in the future. Second, PSM was used to balance potential confounding factors, however, some unmeasured variables might still be there. Third, as we had data from between 2013 and 2015 only, we were able to analyze only data corresponding to this study period. Although the study period did not influence the conclusions, future studies with updated data are warranted.

## 5. Conclusions

Third-generation cephalosporin resistance increased economic costs and prolonged LOS among inpatients with *E. coli* and *K. pneumoniae*. In addition, inpatients with 3GCRKP were significantly associated with higher hospital mortality compared with 3GCSKP cases, however, there was no significant difference in hospital mortality between the 3GCREC and 3GCSEC groups. Given the clinical and economic burden associated with 3GCREC and 3GCRKP that we have demonstrated, efforts to control the development and spread of third-generation cephalosporin resistance and *E. coli* and *K. pneumoniae* should be a priority. Cost reduction and outcome improvement could be achieved through a preventative approach in terms of both antimicrobial stewardship and preventing the transmission of organisms. In addition, proper assessment before the empirical use of third-generation cephalosporins is recommended to mitigate costs.

## Figures and Tables

**Table 1 ijerph-17-09285-t001:** Characteristics of patients with 3GCREC and 3GCSEC and with 3GCRKP and 3GCSKP before PSM and after PSM.

	Before PSM						After PSM for Potential Confounding Variables Excluding LOS Before Culture						After PSM for Potential Confounding Variables Including LOS Before Culture					
Baseline Characteristics	3GCSEC	3GCREC	*p* Value	3GCSKP	3GCRKP	*P* Value	3GCSEC	3GCREC	*p* Value	3GCSKP	3GCRKP	*p* Value	3GCSEC	3GCREC	*P* Value	3GCSKP	3GCRKP	*p* Value
Number of inpatient, *n*	2588	2056		3485	1679		1815	1815		1617	1617		1804	1804		1521	1521	
Age in years, median (range)	73 (0–100)	72 (0–100)	0.196	72 (0–100)	74 (0–99)	<0.000	72 (0–100)	72 (0–100)	0.233	71 (0–100)	74 (0–99)	0.465	72 (0–100)	73 (0–100)	0.843	73 (0–99)	70 (0–99)	0.396
Sex male, *n* (%)	1174 (45.36)	600 (29.18)	<0.000	2357 (67.63)	1163 (69.27)	0.238	585 (32.23)	600 (33.06)	0.595	1123 (69.45)	1113 (68.83)	0.703	600 (33.26)	582 (32.26)	0.523	1052 (69.17)	1072 (70.48)	0.43
Insurance, *n* (%)	2262 (87.40)	1799 (87.50)	0.921	2859 (82.04)	1374 (81.83)	0.859	1583 (87.22)	1590 (87.60)	0.726	1305 (80.71)	1320 (81.63)	0.5	1582 (87.69)	1567 (86.86)	0.454	1241 (81.59)	1224 (80.47)	0.432
Number of diagnoses, median (range)	6 (1–23)	6 (1–20)	0.452	6 (1–30)	7 (1–23)	<0.000	6 (1–23)	6 (1–20)	0.87	7 (1–30)	6 (1–21)	0.353	6 (1–20)	6 (1–23)	0.753	7 (1–23)	6 (1–30)	0.687
Charlson comorbidity index, median (range)	5 (1–29)	5 (1–37)	0.654	5 (1–34)	5 (1–30)	0.603	5 (1–29)	5 (1–37)	0.722	5 (1–34)	5 (1–30)	0.654	5 (1–37)	5 (1–28)	0.902	5 (1–30)	5 (1–27)	0.180
Admission to ICU, *n* (%)	175 (6.76)	87 (4.23)	<0.000	420 (12.05)	404 (24.06)	<0.000	88 (4.85)	87 (4.79)	0.938	348 (21.52)	357 (22.08)	0.701	87 (4.82)	90 (4.99)	0.817	317 (20.84)	332 (21.83)	0.507
Surgery, *n* (%)	770 (29.75)	451 (21.94)	<0.000	868 (24.91)	514 (30.61)	<0.000	464 (25.56)	447 (24.63)	0.515	498 (30.80)	490 (30.30)	0.76	448 (24.83)	445 (24.67)	0.908	459 (30.18)	496 (32.61)	0.148
Myocardial infarction, *n* (%)	63 (2.43)	47 (2.29)	0.741	102 (2.93)	41 (2.44)	0.32	44 (2.42)	43 (2.37)	0.914	40 (2.47)	41 (2.54)	0.91	44 (2.44)	37 (2.05)	0.431	34 (2.24)	41 (2.70)	0.413
Congestive heart failure, *n* (%)	439 (16.96)	296 (14.40)	0.017	627 (17.99)	258 (15.37)	0.019	283 (15.59)	293 (16.14)	0.65	229 (14.16)	248 (15.34)	0.346	296 (16.41)	279 (15.47)	0.439	235 (15.45)	220 (14.46)	0.446
Peripheral vascular disease, *n* (%)	19 (0.73)	14 (0.68)	0.83	47 (1.35)	18 (1.07)	0.404	13 (0.72)	14 (0.77)	0.847	18 (1.11)	18 (1.11)	1	14 (0.78)	18 (1.00)	0.478	18 (1.18)	21 (1.38)	0.629
Cerebrovascular diseases, *n* (%)	1077 (41.62)	960 (46.69)	0.001	1786 (51.25)	881 (52.47)	0.41	783 (43.14)	813 (44.79)	0.316	840 (51.95)	845 (52.26)	0.86	794 (44.01)	785 (43.51)	0.763	790 (51.94)	787 (51.74)	0.913
Dementia, *n* (%)	91 (3.52)	74 (3.60)	0.879	81 (2.32)	73 (4.35)	<0.000	62 (3.42)	68 (3.75)	0.592	69 (4.27)	67 (4.14)	0.861	67 (3.71)	67 (3.71)	1	61 (4.01)	65 (4.27)	0.716
Chronic pulmonary disease, *n* (%)	442 (17.08)	261 (12.69)	<0.000	891 (25.57)	351 (20.91)	<0.000	266 (14.66)	259 (14.27)	0.741	328 (20.28)	344 (21.27)	0.488	260 (14.41)	264 (14.63)	0.85	323 (21.24)	303 (19.92)	0.37
Connective tissue disease, *n* (%)	84 (3.25)	88 (4.28)	0.064	62 (1.78)	29 (1.73)	0.894	73 (4.02)	74 (4.08)	0.933	30 (1.86)	28 (1.73)	0.791	77 (4.27)	73 (4.05)	0.739	28 (1.84)	29 (1.91)	0.894
Mild liver disease, *n* (%)	121 (4.68)	114 (5.54)	0.179	124 (3.56)	65 (3.87)	0.574	92 (5.07)	88 (4.85)	0.76	63 (3.90)	64 (3.96)	0.928	91 (5.04)	94 (5.21)	0.821	58 (3.81)	67 (4.40)	0.411
Peptic ulcer disease, *n* (%)	62 (2.40)	57 (2.77)	0.42	105 (3.01)	55 (3.28)	0.61	48 (2.64)	45 (2.48)	0.753	53 (3.28)	52 (3.22)	0.921	48 (2.66)	49 (2.72)	0.918	49 (3.22)	53 (3.48)	0.687
Diabetes mellitus, *n* (%)	894 (34.54)	706 (34.34)	0.884	952 (27.32)	448 (26.68)	0.631	628 (34.60)	630 (34.71)	0.944	401 (24.80)	434 (26.84)	0.185	633 (35.09)	625 (34.65)	0.78	411 (27.02)	409 (26.89)	0.935
Diabetes mellitus with chronic complications, *n* (%)	132 (5.10)	167 (8.12)	<0.000	115 (3.30)	63 (3.75)	0.404	119 (6.56)	93 (5.12)	0.066	59 (3.65)	61 (3.77)	0.852	97 (5.38)	121 (6.71)	0.094	57 (3.75)	55 (3.62)	0.847
Moderate to severe chronic kidney disease, *n* (%)	232 (8.96)	188 (9.14)	0.832	235 (6.74)	189 (11.26)	<0.000	171 (9.42)	166 (9.15)	0.775	176 (10.88)	165 (10.20)	0.529	166 (9.20)	167 (9.26)	0.954	153 (10.06)	165 (10.85)	0.477
Hemiplegia, *n* (%)	33 (1.28)	22 (1.07)	0.521	24 (0.69)	22 (1.31)	0.026	18 (0.99)	21 (1.16)	0.629	24 (1.48)	20 (1.24)	0.544	21 (1.16)	19 (1.05)	0.75	18 (1.18)	20 (1.31)	0.744
Solid tumor without metastases, *n* (%)	316 (12.21)	207 (10.07)	0.022	224 (6.43)	126 (7.50)	0.149	198 (10.91)	204 (11.24)	0.751	128 (7.92)	124 (7.67)	0.793	206 (11.42)	197 (10.92)	0.634	121 (7.96)	126 (8.28)	0.74
Leukemia, *n* (%)	40 (1.55)	21 (1.02)	0.119	51 (1.46)	40 (2.38)	0.019	22 (1.21)	21 (1.16)	0.878	43 (2.66)	38 (2.35)	0.574	21 (1.16)	19 (1.05)	0.75	35 (2.30)	39 (2.56)	0.638
Malignant lymphoma, *n* (%)	34 (1.31)	12 (0.58)	0.013	33 (0.95)	28 (1.67)	0.025	8 (0.44)	12 (0.66)	0.37	27 (1.67)	25 (1.55)	0.78	12 (0.67)	13 (0.72)	0.841	24 (1.58)	30 (1.97)	0.41
Severe liver disease, *n* (%)	52 (2.01)	33 (1.61)	0.307	45 (1.29)	26 (1.55)	0.457	29 (1.60)	32 (1.76)	0.698	27 (1.67)	26 (1.61)	0.89	32 (1.77)	33 (1.83)	0.9	23 (1.51)	23 (1.51)	1.000
Metastatic tumor, *n* (%)	129 (4.98)	112 (5.45)	0.48	206 (5.91)	59 (3.51)	<0.000	99 (5.45)	88 (4.85)	0.409	70 (4.33)	59 (3.65)	0.323	91 (5.04)	97 (5.38)	0.653	58 (3.81)	40 (2.63)	0.065

3GCREC: third-generation cephalosporin-resistant *Escherichia coli*; 3GCSEC: third-generation cephalosporin-susceptible *E. coli*; 3GCSKP: third-generation cephalosporin-resistant *Klebsiella pneumoniae*; PSM: propensity score matching; LOS: length of stay; ICU: intensive care unit.

**Table 2 ijerph-17-09285-t002:** Economic costs of patients with 3GCREC and 3GCSEC for potential confounding variables.

Confounding Variables	Hospital Cost ($)	3GCSEC			3GCREC			Difference			*p* Value
		Median	95% CI		Median	95% CI		Median	95% CI		
Excluding LOS before culture	Total hospital cost	3867	3558	4185	5233	4737	5638	1366	1179	1453	<0.000
	Antibiotic cost	126	99	143	278	246	311	152	146	168	<0.000
	Medication cost	1418	1286	1563	2045	1863	2279	627	577	715	<0.000
	Diagnostic cost	873	844	914	955	901	992	81	57	79	<0.000
	Treatment cost	778	719	858	1142	1043	1250	363	324	393	<0.000
	Material cost	187	160	225	321	289	368	134	129	143	<0.000
	Other costs	8	8	9	9	8	10	1	0	1	0.003
Including LOS before culture	Total hospital cost	4057	3791	4435	5197	4733	5662	1140	942	1227	<0.000
	Antibiotic cost	132	108	150	260	235	297	127	127	147	<0.000
	Medication cost	1522	1385	1689	2037	1840	2281	515	456	592	<0.000
	Diagnostic cost	886	848	916	953	909	1002	67	61	85	<0.000
	Treatment cost	841	773	934	1111	1018	712	271	245	296	<0.000
	Material cost	199	172	238	306	273	352	107	101	114	<0.000
	Other costs	8	7	9	9	8	10	1	1	1	0.0213

3GCREC: third-generation cephalosporin-resistant *Escherichia coli*; 3GCSEC: third-generation cephalosporin-susceptible *E. coli*; LOS: length of stay; CI: certainty interval.

**Table 3 ijerph-17-09285-t003:** Economic costs of patients with 3GCRKP and 3GCSKP for potential confounding variables.

Potential Confounding Variables	Hospital Cost ($)	3GCSKP			3GCRKP			Difference			*p* Value
		Median	95% CI		Median	95% CI		Median	95% CI		
Excluding LOS before culture	Total hospital cost	8084	7380	9029	15,754	14,799	16,961	7671	7419	7932	<0.000
	Antibiotic cost	490	430	538	1372	1239	1521	881	809	982	<0.000
	Medication cost	3461	3122	3781	7923	7290	8439	4461	4168	4658	<0.000
	Diagnostic cost	1397	1332	1472	2017	1898	2180	620	566	708	<0.000
	Treatment cost	1637	1491	1768	3249	2992	3524	1612	1501	1756	<0.000
	Material cost	472	419	536	1055	954	1177	583	535	641	<0.000
	Other costs	14	12	16	17	15	20	3	3	4	0.079
Including LOS before culture	Total hospital cost	9699	9089	10,537	14,463	13,428	15,561	4763	4340	5025	<0.000
	Antibiotic cost	526	467	590	1255	1122	1404	729	655	814	<0.000
	Medication cost	4166	3811	4571	7164	6506	7881	2998	2695	3310	<0.000
	Diagnostic cost	1452	1380	1554	1896	1761	2014	445	380	460	<0.000
	Treatment cost	2043	1831	2240	2995	2820	3255	952	989	1015	<0.000
	Material cost	623	566	689	963	866	1071	340	299	383	<0.000
	Other costs	16	14	18	16	14	19	0	1	1	0.4680

3GCRKP: third-generation cephalosporin-resistant *Klebsiella pneumoniae*; 3GCSKP: third-generation cephalosporin-susceptible *K. pneumoniae*; LOS: length of stay; CI: certainty interval.

**Table 4 ijerph-17-09285-t004:** Length of stay of patients with 3GCREC and 3GCSEC and with 3GCRKP and 3GCSKP for potential confounding variables.

Potential Confounding Variables	LOS (Days)	Third-Generation Cephalosporins-Susceptible			Third-Generation Cephalosporins-Resistant			Difference			*p* Value
		Median	95% CI		Median	95% CI		Median	95% CI		
Excluding LOS before culture	3GCREC vs. 3GCSEC	16	16	17	20	19	21	4	3	4	<0.000
	3GCRKP vs. 3GCSKP	20	19	21	31	30	32	11	11	11	<0.000
Including LOS before culture	3GCREC vs. 3GCSEC	17	16	17	19.5	18	21	2.5	2	4	<0.000
	3GCRKP vs. 3GCSKP	23	22	24	30	29	31	7	7	7	<0.000

3GCREC: third-generation cephalosporin-resistant *Escherichia coli*; 3GCSEC: third-generation cephalosporin-susceptible *E. coli*; 3GCRKP: third-generation cephalosporin-resistant *Klebsiella pneumoniae*; 3GCSKP: third-generation cephalosporin-susceptible *K. pneumoniae*; LOS: length of stay; CI: certainty interval.

**Table 5 ijerph-17-09285-t005:** Hospital mortality of patients with 3GCREC and 3GCSEC and with 3GCRKP and 3GCSKP for potential confounding variables.

Potential Confounding Variables	Mortality Rate (%)	Third-Generation Cephalosporins-Susceptible			Third-Generation Cephalosporins-Resistant			Difference			*p* Value
		Rate	95% CI		Rate	95% CI		Rate	95% CI		
Excluding LOS before culture	3GCREC vs. 3GCSEC	2.15	1.58	2.93	2.7	2.05	3.55	0.55	0.47	0.62	0.281
	3GCRKP vs. 3GCSKP	3.65	2.84	4.68	6.74	5.62	8.07	3.09	2.78	3.39	<0.000
Including LOS before culture	3GCREC vs. 3GCSEC	2.16	1.58	2.94	2.49	1.87	3.32	0.33	0.29	0.38	0.508
	3GCRKP vs. 3GCSKP	3.81	2.96	4.89	6.51	5.35	7.9	2.7	2.39	3.01	0.001

3GCREC: third-generation cephalosporin-resistant *Escherichia coli*; 3GCSEC: third-generation cephalosporin-susceptible *E. coli*; 3GCRKP: third-generation cephalosporin-resistant *Klebsiella pneumoniae*; 3GCSKP: third-generation cephalosporin-susceptible *K. pneumoniae*; LOS: length of stay; CI: certainty interval.

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
