# Peer review of "Clinical and Economic Impact of Third-Generation Cephalosporin-Resistant Infection or Colonization Caused by Escherichia coli and Klebsiella pneumoniae: A Multicenter Study in China"

_ijerph, 2020, doi:10.3390/ijerph17249285_

Round 1

Author Response

The manuscript by Zhen et al, titled “Clinical and economic impact of third-generation 2 cephalosporin-resistant infection or colonization 3 caused by Escherichia coli and Klebsiella pneumoniae: 4 a multicenter study in China” is providing an important analysis of public health related data as a basic reference for further study in the field. The outcome of this study has a geographical importance to promote basic and clinical research on cephalosporin-resistant infections. However, the manuscript is lack of a strong argument to emphasize the importance of their study. The manuscript more focused on explaining the statistical analysis but poor focus in related to public health importance and/or antimicrobial stewardship. Thus, this may be less appealing as a mean of scientific communication to the public health experts. Please respond to the following comments/suggestions to improve the quality of this study.

Overall, the writing style need to be improved for smoother flow. Too many uses of “It” – that has no connection to former or latter sentences.

Abstract:

Abstract was not written according to the standard format. Introduction is missing. Abbreviations are not properly defined, for example, 3GCREC. Too many of results are described in the abstract so please concise to major findings to keep it in the flow and interesting to read. Rather than saying “It was a retrospective and multicenter study”, it would be more clearer “…using a retrospective and multicenter study” or use a more proper third person sentence.

Response: Thank you very much for your comments.

  • We have revised the abstract according to the standard format. Please see in the Abstract section, page 2, lines 22-41.
  • We are so sorry for our careless, we have properly defined the abbreviations in the abstract, and defined all abbreviations separately in the revised manuscript. Please see in the Abstract section, page 2, lines 22-41; Abbreviations section, page 14, lines 229-235.
  • We have revised the results in the abstract. The proportions of 3GCREC and 3GCRKP in the sampled hospitals were 44.3% and 32.5%, respectively. In the two PSM methods, there generated 1,804 pairs and 1,521 pairs, and there obtained 1,815 pairs and 1,617 pairs, respectively. Compared with susceptible cases, those with 3GCREC and 3GCRKP were associated with significantly increased total hospital cost and excess LOS. Inpatients with 3GCRKP were significantly associated with higher hospital mortality compared with 3GCSKP cases, however, there was no significant difference between 3GCREC and 3GCSEC cases. Please see in the Abstract section, page 2, lines 34-40.
  • We have revised the saying “It was a retrospective and multicenter study” into “A retrospective and multicenter study was conducted.” Please see in the Abstract section, page 2, line 28.

As this manuscript has too many abbreviations, I suggest you to define abbreviations separately.

Response: Thank you very much for your comments.

We have defined all abbreviations separately in the revised manuscript. Please see in Abbreviations section, page 14, lines 229-235.

Introduction:

Line 40-44: This introduction sentence should be supported by more than one reference to highlight the importance and widespread of the problem.

Response: Thank you very much for your comments.

We have added other references to highlight the importance and widespread of the problem. “Olalekan A, Onwugamba F, Iwalokun B, Mellmann A, Becker K, Schaumburg F. High proportion of carbapenemase-producing Escherichia coli and Klebsiella pneumoniae among extended-spectrum beta-lactamase-producers in Nigerian hospitals. Journal of Global Antimicrobial Resistance. 2020, 21:8-12.” and “Pitout JDD, Laupland KB. Extended-spectrum beta-lactamase-producing enterobacteriaceae: an emerging public-health concern. The Lancet Infectious Diseases. 2008, 8:159-166.”.

Please see in the Introduction section, page 3, lines 47-52.

Line 52 and 54; what is ‘it’ refer to? Better starting words would be “….et al showed/pointed out” or In a study by/on ..”

Response: Thank you very much for your comments. We have revised the manuscript accordingly. Please see in the Introduction section, page 3, lines 53-57.

Line 59-60: Highlight the importance/need for such a study in China rather than just saying ‘uninvestigated’. Why this investigation is necessary?

Response: Thank you very much for your comments. We have revised the manuscript accordingly.

Quantifying clinical and economic outcomes would facilitate strategies towards for containment of third-generation cephalosporin resistance and E. coli or K. pneumoniae. Resistance to third-generation cephalosporins by E. coli or K. pneumoniae, which represented the major mechanism of antimicrobial resistance, had been reported as independently associated with a poor outcome and increased use of healthcare resources. However, no significant difference in hospital mortality between 3GCREC and 3GCSEC was reported as well. In China, there was only one study exploring longer LOS and higher hospital costs attributable to extended spectrum beta-lactamase-positive intra-abdominal infection caused by E. coli or K. pneumoniae. The clinical and economic outcomes of 3GCREC and 3GCRKP remained largely uninvestigated in China.

Please see in the Introduction section, page 3, lines 67-76.

Materials & Methods:

2.2: Avoid too many uses of ‘it’.

Response: Thank you very much for your comments. We have revised the manuscript accordingly.

Please see in the Materials and Methods section, page 3, line87; line 90.

Line 73-74. Please explain. This is very contradictory statement as colonize patient will have no clinical sign and no antibiotics are indicated for such scenarios. Does the ref 15 support your methodology? If so how? Re-consider adding the explanation either to the Introduction or Discussion section as appropriate.

Response: Thank you very much for your comments.

  • Colonization of coli and K. pneumoniae, as the reservoir for infection with these organisms, was also a risk factor for higher mortality, longer LOS, and increased hospital costs. Pleases see in the Introduction section, page 3, lines 65-66.
  • We added another reference to support the methodology. “Hamprecht A, Rohde AM, Behnke M, Feihl S, Gastmeier P, Gebhardt F, Kern WV, Knobloch JK, Mischnik A, Obermann B et al. Colonization with third-generation cephalosporin-resistant Enterobacteriaceae on hospital admission: prevalence and risk factors. The Journal of Antimicrobial Chemotherapy. 2016, 71:2957-2963.” Please see in the Materials and Methods section, page 4, line 93.
  • We discussed it in the limitations. Due to the retrospective nature of our study, it was difficult to distinguish infection or colonization. It was necessary to explore the burden of 3GCREC and 3GCRKP, either infection or colonization, because that colonization was an important reservoir for organisms of infection. Prospective studies among patients with infections need to conducted in the future. Pleases see in the Discussion section, page 13, lines 210-213.

How did you determine the antibiotics sensitivity? Did you use any automated instrument or Kirby-bauer? All centers should have a unified AST method and confirmatory ESBL and AmpC testing strategy to categorize as you mentioned here. Please provide more details on your bioassays.

Response: We determined the antibiotic sensitivity according to Clinical and Laboratory Standards Institute definition.

Please see in the Materials and Methods section, pages 3-4, lines 90-93.

Line 89: What is PSM? Explain for non-statistic readership (in addition to abstract for clarity). It could be better to have a separate section for “Abbreviations” as there are lot of abbreviations throughout the writing. You need to explain why you selected this statics method.

Response: Thank you very much for your comments.

  • Propensity score matching (PSM), widely used to control for confounding in observational studies, is a powerful statistical matching technique for reducing a set of confounding variables to a single propensity score in order to effectively control for all observed confounding bias. Please see in the Materials and Methods section, page4, lines 103-105.
  • We have defined all abbreviations separately in the revised manuscript. Please see in Abbreviations section, page 14, lines 229-235.

Results

Result section should be start stating the prevalence of ESBL in all centers you collected data before analyzing the data. That would give a neat flow to guide the reader.

Response: Thank you very much for your comments. We have revised the manuscript accordingly.

The proportions of 3GCREC and 3GCRKP in the sampled hospitals were 44.3% and 32.5%, respectively.

Please see in the Results section, page 4, lines 124.

Line 103: Colonize is not an indication of antibiotics therapy. How did you establish this?

Response: Thank you very much for your comments.

  • Colonization of coli and K. pneumoniae, as the reservoir for infection with these organisms, was also a risk factor for higher mortality, longer LOS, and increased hospital costs. Pleases see in the Introduction section, page 3, lines 65-66.
  • We discussed it in the limitations. Due to the retrospective nature of our study, it was difficult to distinguish infection or colonization. It was necessary to explore the burden of 3GCREC and 3GCRKP, either infection or colonization, because that colonization was an important reservoir for organisms of infection. Prospective studies among patients with infections need to conducted in the future. Pleases see in the Discussion section, page 13, lines 210-213.

The authors should explain the outcomes of the data Table 1-4 and the summary rather than just showing up a table with no explanation. In scientific publications, summarizing a table/figure in the text will enhance the readers’ understanding of the study.

Response: Thank you very much for your comments. We have revised the manuscript accordingly.

Table 1-3 described the characteristics of patients with 3GCREC and 3GCSEC and with 3GCRKP and 3GCSKP before PSM and after PSM.

Table 4-5 showed economic costs of patients with 3GCREC and 3GCSEC and with 3GCRKP and 3GCSKP after PSM.

Table 6-7 showed length of stay and hospital mortality of patients with 3GCREC and 3GCSEC and with 3GCRKP and 3GCSKP after PSM, respectively.

Please see in the Results section, pages 4-8, lines123-174; Table 1-7.

Discussion

Please clarify the paragraph 2 (Line 153-161) with better flow to improve the readability. Line 153-154: need references. Avoid using ‘It” with no clue. What is meant by “The conclusions have been revealed before”?

Response: Thank you very much for your comments.

  • We have revised the paragraph 2 in the discussion. Please see in the Discussion section, page 13, lines 186-194.
  • We have added the references. Please see in the Discussion section, page 13, line 189.
  • “The conclusions have been revealed before” had been deleted.

Line 162: What is the link given here saying “At the same time”?

Response: We have revised the saying “At the same time” into “In addition”. Please see in the Discussion section, page 13, line 195.

Line 162 – 167: Are these differences are due to different study methods or different genetic makeup in European population? in Line 166-167: Do you expect to see the same pattern in Chinese population or are there any potential environmental factors to be affect for this requirement? Justify/validate your statement. Could be a better to move this argument/explanation to the introduction part.

Response: Thank you very much for your comments.

  • Colonization of coli and K. pneumoniae, as the reservoir for infection with these organisms, was also a risk factor for higher mortality, longer LOS, and increased hospital costs. Pleases see in the Introduction section, page 3, lines 65-66.
  • Different conclusions might be associated with different study design, sample size, geography, resistant pattern, etc. Therefore, this finding needs to be further explored in the future. Please see in the Discussion section, page 13, lines 200-202.

Line 185: This section should have a recommendation to avoid higher cost of drug use.

Response: Thank you very much for your comments.

Cost reduction and outcome improvement could be achieved through preventative approach in both antimicrobial stewardship and preventing transmission of organisms. In addition, proper assessment before empirical use of third-generation cephalosporins were recommended to mitigate costs. Please see in the Conclusions section, page 14, lines 225-227.

Conclusions:

This section should mention what to implement to mitigate cost. Why don't you use proper AST assessment before empirical use of third generation cephalosporins? If that’s is shortcoming, need to list as a limitation to the study.

Response: Thank you very much for your comments.

Cost reduction and outcome improvement could be achieved through preventative approach in both antimicrobial stewardship and preventing transmission of organisms. In addition, proper assessment before empirical use of third-generation cephalosporins were recommended to mitigate costs. Please see in the Conclusions section, page 14, lines 225-227.

Reviewer 2 Report

Dear authors,

This type of work helps considerably to use not only antibiotics rationally, but also the resources available for treatments. Since the economic return to the pharmaceutical industry is scarce.

Author Response

This type of work helps considerably to use not only antibiotics rationally, but also the resources available for treatments. Since the economic return to the pharmaceutical industry is scarce.

Response: Thank you very much for your comments.

Reviewer 3 Report

The manuscript “Clinical and economic impact of third-generation cephalosporin-resistant infection or colonization caused by Escherichia coli and Klebsiella pneumoniae: a multicenter study in China” presents an interesting study on clinical and economic outcomes between third-generation cephalosporin-resistant E. coli (3GCREC) and third-generation cephalosporin-susceptible Escherichia coli (3GCSEC) cases, and between cephalosporin-resistant K. pneumonia (3GCRKP) and third-generation cephalosporin-susceptible Klebsiella pneumoniae (3GCSKP) cases. This is a retrospective study, so it has limitation on data exploration, though this study illustrates significance several clinical impact on resistant and susceptible bacterial infections.

  1. The extension of 3GCREC and 3GCRKP must be included at first use in abstract.
  2. Discussion part lacks conclusive remarks on economic impacts.
  3. Some grammatical and spelling errors.

Author Response

The manuscript “Clinical and economic impact of third-generation cephalosporin-resistant infection or colonization caused by Escherichia coli and Klebsiella pneumoniae: a multicenter study in China” presents an interesting study on clinical and economic outcomes between third-generation cephalosporin-resistant E. coli (3GCREC) and third-generation cephalosporin-susceptible Escherichia coli (3GCSEC) cases, and between cephalosporin-resistant K. pneumonia (3GCRKP) and third-generation cephalosporin-susceptible Klebsiella pneumoniae (3GCSKP) cases. This is a retrospective study, so it has limitation on data exploration, though this study illustrates significance several clinical impact on resistant and susceptible bacterial infections.

The extension of 3GCREC and 3GCRKP must be included at first use in abstract.

Response: We are so sorry for our careless, we have properly defined the abbreviations in the abstract, and defined all abbreviations separately in the revised manuscript. Please see in the Abstract section, page 2, lines 22-41; Abbreviations section, page 14, lines 229-235.

Discussion part lacks conclusive remarks on economic impacts.

Response: Thank you very much for comments.

Previous studies mainly focused on antibiotic utilization and resistance mechanism and the clinical and economic outcomes of 3GCREC and 3GCRKP in China remained largely uninvestigated. To the best of our knowledge, this is the first study to quantify the clinical and economic outcome of 3GCREC and 3GCRKP in mainland China using the PSM method with large sample size and multiple hospital settings. We focused on E. coli or K. pneumoniae, avoiding non-specific effect from a combination of bacteria. In this study, we found that compared with third-generation cephalosporin-susceptible cases, those with 3GCREC and 3GCRKP were associated with significantly increased total hospital cost and excess LOS. In addition, inpatients with 3GCRKP were significantly associated with higher hospital mortality compared with 3GCSKP cases, however, there was no significant difference between 3GCREC and 3GCSEC group.

Please see in the Discussion section, page 13, lines 175-185.

Some grammatical and spelling errors.

Response: Thank you very much for your comments, We have revised the grammatical and spelling errors throughout the manuscript. Please see the revised manuscript.

Round 2

Reviewer 1 Report

After a major revisions in clarifying the contents and text editing, the manuscript by Zhen et al, titled “Clinical and economic impact of third-generation 2 cephalosporin-resistant infection or colonization 3 caused by Escherichia coli and Klebsiella pneumoniae: a multicenter study in China” appears in a readability format. Thank you for the extensive efforts in revising.

Following minor points are noteworthy.

1. When dealing with a work by three to five authors in referencing within the text, use the first author's last name in the signal phrase or parentheses, followed by et al. For example:

  • Lucas et al. (1995) explores...
  • (Lucas et al., 1995)

Accordingly, correct the line 47, 54 etc. throughout (last name only).

2. Check whether funding information are sufficiently declared according to the standard format. No inverted commas are needed. Refer to few other published journal articles from IJERPH.

3. Table 1, 2 and 3 could be adjusted to fit one page.

Author Response

After a major revisions in clarifying the contents and text editing, the manuscript by Zhen et al, titled “Clinical and economic impact of third-generation 2 cephalosporin-resistant infection or colonization 3 caused by Escherichia coli and Klebsiella pneumoniae: a multicenter study in China” appears in a readability format. Thank you for the extensive efforts in revising.

Following minor points are noteworthy.

  1. When dealing with a work by three to five authors in referencing within the text, use the first author's last name in the signal phrase or parentheses, followed by et al. For example:

Lucas et al. (1995) explores...

(Lucas et al., 1995)

Accordingly, correct the line 47, 54 etc. throughout (last name only).

Response: Thank you very much for your comments. We have revised the manuscript accordingly.

Please see in the Introduction section, page 3, lines 53, 60, 64; Discussion section, page 10, lines 197, 199.

  1. Check whether funding information are sufficiently declared according to the standard format. No inverted commas are needed. Refer to few other published journal articles from IJERPH.

Response: Thank you very much for your comments. We have referred to few other published journal articles from IJERPH, and revised the funding information according to the standard format.

This work was jointly supported by the Pfizer Investment Co. Ltd (Burden of multi-drug resistant infections in China and associated risk factors), the Fundamental Research Funds of Shandong University, and the Joint Research Funds for Shandong University and Karolinska Institutet.

Please see in the Funding section, page 11, lines 247-250.

  1. Table 1, 2 and 3 could be adjusted to fit one page.

Response: Thank you very much for your comments, we have adjusted Table 1-3 to fit one page.

Please see the Table 1-3 in the revised manuscript.